# Achieving cervical cancer elimination: The simulated impacts of HPV vaccination and transitioning from liquid-based cytology to HPV-based screening test

Nur Zahirah Balqis-Ali[1], Jailani Anis-Syakira[1]*, Weng Hong Fun[1], Zakiah Mohd Said[2], Shazimah Abdul Samad[2], Norhaslinda Zainal Abidin[3,4], Jafri Zulkepli[3], Norazura Ahmad[3,4], Mohd Norazam Mohd Abas[5], Chee Meng Yong[6], Siti Norbayah Yusof[7], Nur Adila Daud[7], Sondi Sararaks[1]

1 Institute for Health Systems Research, National Institutes of Health, Ministry of Health, Centre for Health Outcomes Research, Shah Alam, Selangor, Malaysia, 2 Family Health Section, Family Health Development Division, Ministry of Health, Putrajaya, Malaysia, 3 School of Quantitative Sciences, College of Arts and Sciences, Universiti Utara Malaysia, Sintok, Kedah, Malaysia, 4 Institute of Strategic Industrial Decision Modelling, School of Quantitative Sciences, College of Arts and Sciences, Universiti Utara Malaysia, Sintok, Kedah, Malaysia, 5 Gynae-Oncology Department, National Cancer Institute, Ministry of Health, Putrajaya, Malaysia, 6 Gynae-Oncology Department, Ampang Hospital, Ministry of Health, Selangor, Malaysia, 7 National Cancer Registry, National Cancer Institute, Ministry of Health, Putrajaya, Malaysia

☯ These authors contributed equally to this work.
* anissyakira.j@moh.gov.my

**Data Availability Statement:** The dataset supporting this article's findings belongs to the Family Health Development Division, MOH

## Abstract

The Ministry of Health Malaysia aims to fully replace liquid-based cytology (LBC) with Human Papillomavirus (HPV) tests and increase the screening coverage from a baseline of 25% to 40% by 2023, followed by a 10% yearly increment until 70% coverage. This transition requires proper planning, including the number of tests needed and budget allocation. This study aims to simulate different transition strategies involving the shift from LBC to HPV testing with expanded screening coverage to predict their impact on cervical cancer elimination in Malaysia. A system dynamics model was developed to simulate the transition from LBC to HPV testing and HPV vaccination coverage in Malaysia. The dynamic model utilised local epidemiological data, published research, and expert opinion when data was unavailable. The simulation showed that a combination of high HPV vaccination coverage and transitioning fully to five-yearly HPV testing by 2030, coupled with a 70% screening uptake by 2040, would accelerate cancer elimination in Malaysia by 18 years with an estimated screening cost of MYR1.81 billion equivalent to USD 411 millions compared to the baseline of using LBC as the primary screening method (MYR1.39 billion, USD 315 millions). Sustained, it would lead to averting 6,000 new cancer cases by 2070. Alternatively, conducting HPV screenings twice before age 50 would advance cervical cancer elimination by 14 years and prevent approximately 1000 new cases by 2070, with an estimated cost of MYR1.13 billion equivalent to USD 257 millions. A delay in achieving the full transition prolonged the elimination more than a delay in increasing the screening coverage. In all scenarios, yearly vaccination of 90% of girls age 13 is crucial to achieving elimination. In the

Malaysia and the National Cancer Registry, Malaysia. Requests for the data can be requested through the Principal Investigator (anissyakira. j@moh.gov.my), with permission from the data holders and the Director-General of Health, Malaysia.

**Funding:** Funding for this study is from the Ministry of Health Malaysia research grant [NIH/ 800-3/2/2 Jilid 13(61)]. The funder had no role in the study design, data collection, analysis, decision to publish, preparation or review of the manuscript.

**Competing interests:** The authors declare that they have no competing interests.

**Abbreviations:** DNA, deoxyribonucleic acid; HPV, Human papillomavirus; MOH, Ministry of Health Malaysia; Pap, Papanicolaou test; LBC, Liquid-based cytology; PCR, polymerase chain reaction; WHO, World Health Organization; EY, Elimination year; CC, Cumulative cancer cases; MYR, Malaysian Ringgit.

Malaysian context, where HPV vaccination coverage has reached considerable levels, the evidence advocates for a full transition from LBC to HPV testing, ideally by 2030. While expanding screening coverage remains a critical factor in this endeavour, the findings unequivocally endorse prioritising the transition process.

**Trial registration**: Trial registration number: NMRR ID-22-00187-DJU.

# Introduction

## 1.1 Problem/Significance

The World Health Organization (WHO) calls for global action to eliminate cervical cancer, achieved when cervical cancer incidences fall to four or fewer cases per 100,000 women [1]. The key pillars revolve around prevention through Human Papillomavirus (HPV) vaccination, effective screening, and adequate treatment [1]. In Malaysia, cervical cancer was the third most prevalent cancer among females between 2012 and 2016 [2]. Since 2011, the incidence rate has remained between 6.2 and 7.6 per 100,000 women [2]. Despite having an excellent yearly HPV vaccination program covering around 83–91% of girls aged 13 in the population since 2010 [3], the screening rate of susceptible women in Malaysia remains low and has pla-teaued at about 16–25% over the years, reflected by the plateauing of the cancer incidence [4,5]. Sustained, this could hinder achieving the elimination status; thus, expanding cervical cancer screening becomes the public health priority of Malaysian policy-makers involved in the path towards cervical cancer eradication [6].

The approach towards cervical cancer screening in Malaysia began with the launch of a national screening program using the Papanicolaou (Pap) Smear test in the 1990s. In 2014, Malaysia shifted towards a liquid-based cytology (LBC) test, adhering to its higher sensitivity for abnormal cervical cell detection [7]. The program's opportunistic nature, the need for a national screening registry, and healthcare system-related hurdles are among the challenges in screening more women. Low awareness of cervical cancer and the screening program and fear of performing LBC among Malaysian women further contributed to the low uptake [8,9]. These challenges interfered with the Ministry of Health (MOH) Malaysia's intention to reach 70% screening coverage for at-risk women [6].

To address the issue, Malaysia shifted towards more straightforward and practical testing in screening, HPV DNA-based testing. HPV-based testing provides 60–70% greater protection against invasive cervical carcinomas than LBC due to its ability to detect women at risk at the infectious rather than disease stage, coupled with higher detection sensitivity [10]. Further-more, HPV testing can be done as self-sampling instead of relying on healthcare providers for LBC, further increasing the screening uptake rate [11,12]. The test was made available in phases in healthcare facilities under MOH Malaysia in 2019 [7]. The program targets women aged 30 to 65, with women younger or older encouraged to continue LBC [7]. The ministry aims to fully replace LBC with HPV test and increase the coverage to 40% in 2023, followed by a 10% yearly increment until 70% coverage is achieved [6]. To achieve this, the MOH plans to increase the yearly procurement of HPV test kits to replace LBC and expand screening through collaborative platforms such as private facilities, workplace screening programs, and nonprofit organizational activities [6,13]. However, this transition requires proper planning, including the number of tests needed to be performed and budget allocation since the cost to perform HPV testing is higher than an LBC [14].

Piloting the transition will require significant time and resources before tangible outcomes become evident. However, this vital information supports policymakers to strategise and

allocate budgets effectively, ensuring greater participation of women in screening programs. Thus, a system dynamics simulation, a computer-based modelling and analysis technique addressing the behaviour of complex systems over time [15], was selected to simulate the transition and expansion of HPV testing in Malaysia to provide information assisting policymakers in decision-making. Given the issue's complexity and policy application's goal, this study's approach involved continuous and iterative engagement with key policymakers responsible for implementing Malaysia's screening program to ensure the relevance and robustness of the simulation model.

Although many dynamic models of cervical cancer screening programs have been developed and tested [14,16–18], each model was created to address specific objectives and was applied to different health system contexts. For example, a previous dynamic model projected that eliminating cervical cancer in Malaysia could be accelerated by adopting vaccination and HPV testing as screening tools [14]. This study extends the growing evidence by comparing various transitional scenarios from LBC to HPV tests to accelerate cervical cancer elimination in Malaysia. Such findings are essential for policy-makers in their strategic planning to fully transition from the traditional LBC approach to HPV testing, considering the capacity and other factors shown through the current study.

## 1.2 Aim of the analysis

This study aims to project different transitional scenarios from LBC to HPV testing and increased screening coverage to accelerate cervical cancer elimination in Malaysia. This study seeks to answer the following research questions: First, should the yearly high population coverage of HPV vaccination be sustained at 90%, and what is the impact if the coverage dropped? Second, will the transitions from LBC to HPV testing, in combination with an increment in the screening coverage, accelerate cervical cancer elimination in Malaysia, and by how many years? Finally, given different scenarios in the transition and increment of coverage, what are the impacts on screening costs, the number of tests needed to be performed, and the number of cervical cancers avoided?

## Method

A system dynamics model was developed to simulate the shift from LBC to HPV testing and the extent of HPV vaccination coverage in Malaysia. The model incorporated local epidemiological data, published research, and expert opinions in cases where data was not readily available. Data collection spanned from April 2022 to December 2022, while expert elicitation took place between September 2022 and April 2023. The development of the dynamic model occurred from December 2022 to May 2023.

## 2.1 Justification for utilising system dynamics simulation

System dynamics simulation (SDS) is a computer-based mathematical modelling designed to handle complex, interrelated system analysis [15]. First introduced by Jay W. Forrester in the 1950s, the approach runs multiple interconnected factors and relationships simultaneously, allowing for a comprehensive and whole-system approach rather than an isolated or piecemeal exploration of a policy issue [19]. In the context of the current study, this implies that crucial parameters involved in the cervical cancer development and prevention pathways, including disease progression, population growth, vaccination, screening and treatment interventions, and policy analysis, may all be modelled simultaneously while considering the impact of the interactions between these elements. Furthermore, cervical cancer development varies across different age groups and cancer stages. Coupled with the disparities in deaths, ageing rate, and

policy approach to the various ages, an analysis that can handle these population dynamics and varied policy approaches is necessary [20]. Thus, the current study employs the system dynamics approach to manage numerous interacting disease stages and risks, the interaction between delivery systems and diseased populations, incorporating national policy issues comprehensively and concurrently.

The cervical cancer path involves multiple non-linear and feedback relationships, which can be handled through a system dynamics approach [20,21]. Fig 1 demonstrates this relationship between the precancerous progression and regression, coupled with the impact of vaccination and screening in changing the dynamics of the disease progression. In addition, the passage from contracting persistent HPV infection towards developing cervical cancer takes, on average, about 25 years [22], requiring the model to accommodate this delay between cause and effect [20].

Most importantly, the system dynamics approach considers the accumulation of cases over time, recognising the need to project the incidence and prevalence of diseases such as cervical cancer [23]. This feature allows the introduction of different policy scenarios, as intended by this study, projecting the impact of these scenarios towards the variations and trends of cervical cancer cases in Malaysia in the following decades. Projecting these impacts may assist policy-makers in deciding the most practical approach to achieving the desired outcome.

## 2.2 System dynamics simulation

**2.2.1 Model description.**    Fig 1 describes the disease pathway of cervical cancer, along with interventions preventing the progress of the disease. It was adapted and modified based on existing cervical cancer microsimulation models, including Policy1-Cervix and MISCAN-cervix [24,25]. At a susceptible stage, HPV infections are most commonly acquired through sexual contact [26]. Most sexually active women and men eventually become infected; some may develop persistent infections. In Malaysia, the prevalence of high-risk HPV infection is estimated at 4.33% to 4.75% [13]. Acknowledging that HPV can lead to several types of cancers among both sexes [27], the model was bound and concentrated on HPV infection among females, focusing on cervical cancer disease.

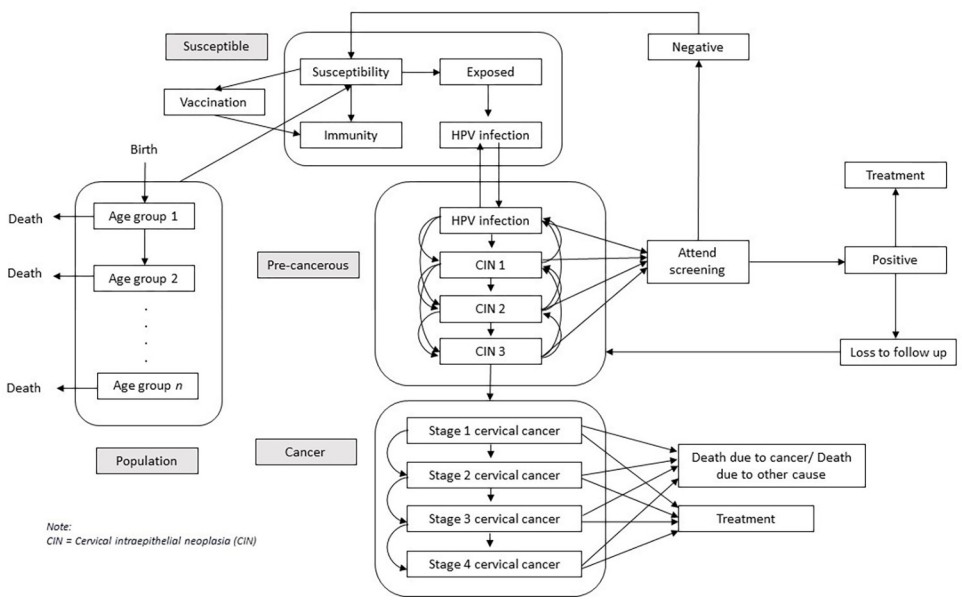

**Fig 1. Disease pathway incorporating population dynamics, vaccination, screening and treatment impact.**

Low-grade cervical abnormalities may be clinically evident in screening with productive HPV infection (low-grade squamous intraepithelial lesions (LSIL) or cervical intraepithelial neoplasia (CIN) grade 1 (CIN1)). They are usually temporary and disappear without intervention within 1–2 years [26]. The majority (90%) of contracted HPV infections typically become untraceable within a span of 1 to 2 years. Left untreated, a small percentage of HPV infections persist for over a year, increasing the likelihood of the development of cervical pre-cancer, such as high-grade squamous intraepithelial lesions (HSIL) or CIN grades 2 or 3 (CIN2/3), and the potential progression to cancer. As shown in the diagram, at this precancerous stage (CIN1 –CIN3), the abnormal cervical cells may either progress or regress spontaneously or intervene through treatment, either progressing a susceptible woman to the cancer stage or regressing to the susceptible pool. However, once it reaches cervical cancer, the regression no longer occurs [26].

Identifying HPV as the causative agent of cervical cancer has resulted in ground-breaking improvements in cervical cancer prevention, including preventive HPV vaccination. HPV vaccination has shown high efficacy and maximal effectiveness [28]. In Malaysia, the vaccine was given only to 13-year-old girls [3]. The quadrivalent vaccine, Gardasil-4 (Merck, Kenilworth, NJ, USA), which protects against HPV genotypes 6, 11, 16, and 18, was modelled since this vaccine was predominantly used in Malaysia [3]. Vaccinated cohorts received lifelong protection against HPV 16 and 18 in the model [29]. This represents 85–91% of the cohort who receives the HPV vaccination yearly [27,30], removed from the cancer pathway. Regarding the screening approach, Malaysia began transitioning from LBC tests to HPV testing in 2019. Besides high sensitivity and specificity, this method will be able to remove the obstacles encountered in the current screening method for cervical cancer [7]. In the model, women attending screening who were detected at the infectious or precancerous stage were assumed to receive optimised treatment or intervention, thus removed from the cancer path. At the same time, those lost to follow-up were returned to the initial disease stage.

The model combined the disease pathway with population dynamics, including declining birth rate and increasing life expectancy. The population began at birth, with the female cohort receiving and protected from the disease path at age 14 (upon completion of both vaccine doses). The population was stratified according to age groups to accommodate differing death, infection, disease progression rates, and policy approaches. The age group classifications are as follows: 0–13, 14 (completed vaccination group), 15–19, 20–29, 30–39, 40–49, 50–65, and above 65. While the diagram in Fig 1 is simplified for comprehensibility, all disease pathways and intervention components were stratified and modelled accordingly by each age group.

**2.2.2 Model assumptions.** The baseline model was simulated based on several assumptions, as follows:

1. *Vaccination assumptions*: Two types of HPV vaccines were approved in the MOH formulary: Cervarix (GlaxoSmithKline, London, UK) and Gardasil-4 (Merck, Kenilworth, NJ, USA), with the vaccine selection reviewed periodically upon purchase contract renewal [3,31]. The efficacy used in the model was based on Gardasil-4 with 100% protection against four HPV genotypes, 16/18/6/11, without cross-protection for other genotypes [29], considering the more extended usage of Gardasil-4. The full-dose vaccination was assumed to provide life-long protection [29]. Two doses of HPV vaccination were given to girls aged 13 with a 6-month interval between doses. Therefore, complete protection and exiting the disease path were assumed to be achieved at age 14. The vaccination coverage was set at 90% yearly population coverage.

2. *Screening assumptions*: Following the national screening guideline, the model simulated screening for women aged 30 to 65 [7]. The screening coverage was assumed to be equally

distributed across different age groups. Pap Smear was introduced into the model in 1998 following the launch of the national screening program and converted to LBC from 2014 onwards. The screening rate was set at 25% in 2015, mimicking actual practice, including three-yearly screenings following a negative LBC result. HPV testing was introduced into the model starting in 2019, using a screen-treat path. The screen-triage-treat was tested in the alternative scenario analysis. The HPV screening cycle was every five years following a negative result, as currently practised [7].

3. *Herd immunity*: Malaysia achieved a high vaccination coverage (90%), contributing to a 100% reduction in HPV prevalence among the female population [32].

4. *Susceptibility to HPV infection*: The susceptibility rate was based on National Health Morbidity Survey (NHMS) data from 2011–2019, whereby about 76–80% of women who were ever married or cohabiting with a partner were assumed to be at risk [33–35].

5. *Adherence to follow-up following positive screening*: Assumed to be at a rate of 80 to 90%. Lower commitment to follow-up was simulated as an alternative scenario.

6. *Screening costs*: MYR 58.66 (USD13.33) per LBC test and MYR 102.86 (USD23.38) per HPV test done through in-house laboratory processing. MYR 84.55 (USD19.22) per HPV test done through outsourcing the laboratory processing was done as an alternative scenario. The details in the cost calculation are reported elsewhere [36]. The monetary figures in this study were denominated in the local currency unit (MYR) and converted to USD, utilising the World Bank's official exchange rate for 2022 (1 USD equivalent to 4.4 MYR) [37]. Constant screening costs without discounted rates were applied in the simulation.

7. *Disease pathway*: The transition rates were averaged across all high-risk HPV genotypes.

**2.2.3 Data preparation.**   Whenever possible, Malaysian secondary data was used in the model. Data on the prevalence of HPV infection between 2019–2021 was analysed from the Malaysian cervical cancer screening database by the Family Health Development Division, MOH Malaysia (BPKK), published elsewhere [13]. Information on cervical cancer incidence and stratification by age group was extracted from the Malaysian National Cancer Registry Report 2012–2016 [2] with additional data provision by the Department of National Cancer Registry for 2017–2018. The impact of treatment and survival rate at different cancer stages were based on the Malaysian Study on Cancer Survival (MySCan) 2018 report [38]. Vaccination data from 2010–2021 was from the Family Health Development Division, MOH Malaysia, while population data was obtained from the Department of Statistics Malaysia (DOSM) [39]. Data on women susceptible to HPV infection was derived from the National Health and Morbidity Survey (NHMS) 2011–2019 [33–35]. Permission and access to use otherwise not publicly available data was obtained. Most disease progression and regression rates were extracted from various evidence in the literature. Other data for Malaysia was estimated by a group of experts consisting of two public health specialists, two obstetrics and gynaecologists, one medical officer, two nurses, two cervical cancer data custodians, and one academician (expert in cervical cancer studies). Details on the list of data sources can be referred to in Appendix A in S1 File.

**2.2.4 Model development.**   The system dynamics model was built using Vensim Professional version 9.3.4 (Ventana, 2022). Fig 2 represents the stock-flow diagram of the model based on the conceptual illustration described earlier, segregated into different sectors for comprehensibility.

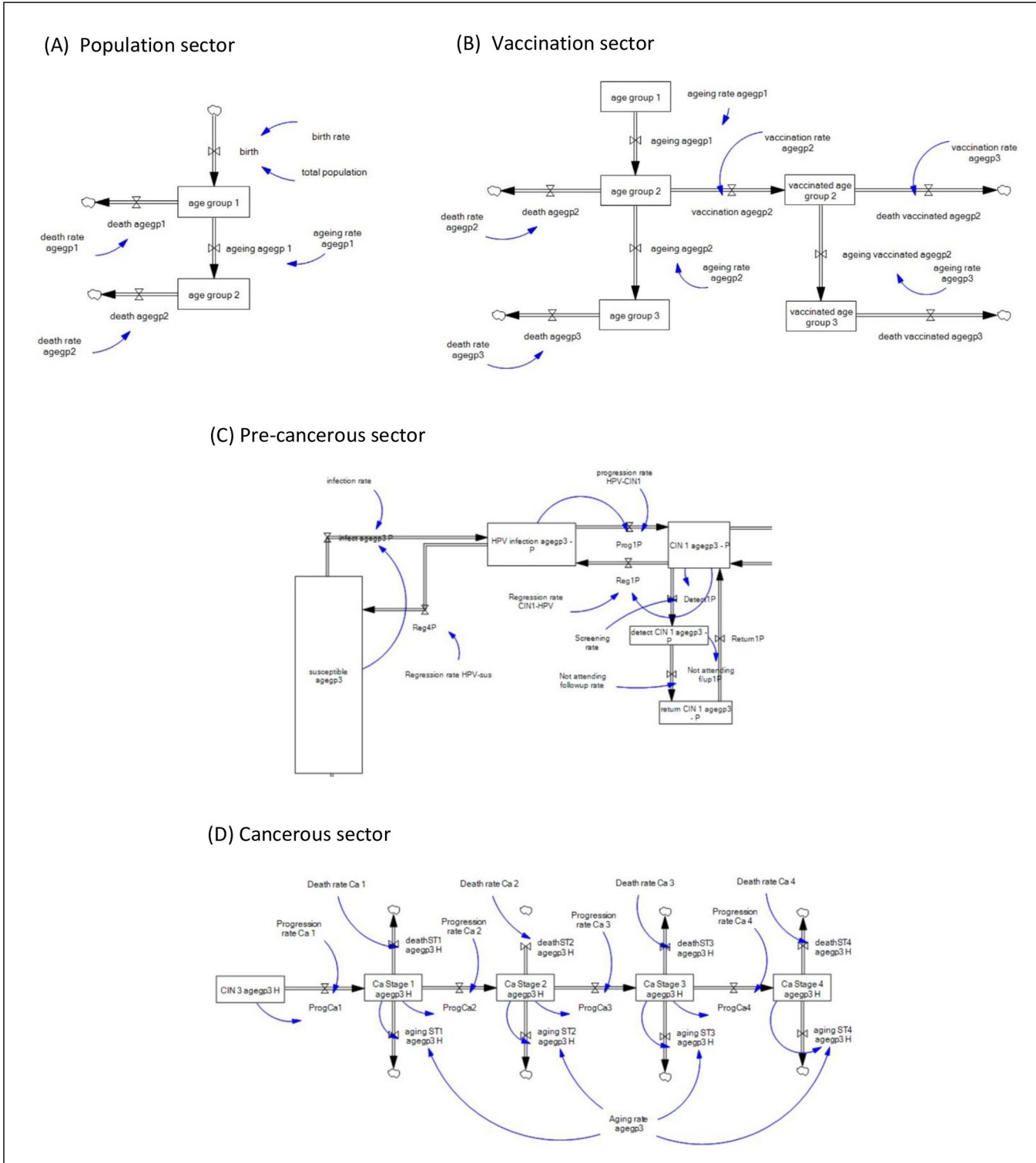

**Fig 2. Stock-flow models of the Malaysian cervical cancer progression, segregated according to the different parts of the model.**

The primary differential equations of integral calculus approximations were used in the system dynamics model as shown in Eq (1) below:

$$Stock(t) \int_{t_0}^{t} (inflow - outflow) * dt + Stock(t_0) \tag{Eq 1}$$

Where $t$ represents the stock, $t_0$ represents the initial value of the stock, and $dt$ is a time step that represents the rate of change concerning time [15].

The stock variables are state variables, illustrated as rectangles in Fig 2 [15]. It accumulates the number of women at various stages in the model by balancing between those entering and leaving the stocks. The inputs and outputs of stocks are linked by flow variables, which indicate rate variability, shown as double-lined arrows. Single-lined arrows show interrelationships between variables. Stock variables include women population from birth till the oldest age group, vaccinated cohort, precancerous and cancerous stages, and cancer incidence. Some of the main equations in the model are shown as follows.

Women from different age groups in the population who were exposed to the infection were calculated as in Eq 2:

*Women exposed to disease*
*= Cohort from previous age group + cohort regressed from HPV infection*
  *+ cohort regressed from CIN 1 + cohort regressed from CIN 2*
  *+ cohort regressed from CIN 3 − cohort contracted HPV infection*
  *− mortality from any cause − cohort ageing to the next age group* (Eq 2)

Women infected with HPV were expressed in Eq 3 as:

*Infected women = Women exposed × rate of infection × susceptibility rate* (Eq 3)

The cohort receiving HPV vaccination was expressed in Eq 4 as:

*Vaccinated equation = Cohort of girls age* 14 × *Vaccine efficacy × Vaccination coverage* (Eq 4)

Women screened every year were calculated in Eq 5 as follows:

*Women attending screening*
  *= (Women exposed + infected + at precancerous stage) × percentage attending screening*
  *× sensitivity of testing* (Eq 5)

Women progressing to cervical cancer were calculated in Eq 6 as follows:

*Women progressing to cervical cancer = Women at CIN3 stage × Rate progressing to cancer* (Eq 6)

Screening at the infectious and precancerous stages was drawn as stocks in the model to allow women successfully detected and treated for having either the infection or precancerous cells to leave the path progressing towards cancer development.

Auxiliary variables are intermediate factors whose values represent significant effects and are calculated by the model [15]. However, it does not capture the cumulative population figures for that specific state; instead, it reflects the count at one point in time. In the model, susceptible women attending screening but found negative were drawn as auxiliaries as the women did not leave the original stock, but need to be calculated to estimate the total number of screening tests needed to be performed. Other auxiliaries include the prevalence of HPV infection, the total number of screening tests performed, cancer staging, and the number of deaths due to cancer. The constants include infectious rates, progression and regression rates at the precancerous stage, progression rate at the cancerous stage, follow-up adherence rate,

sensitivity detection rate of screening tools, screening coverage, screening rate, vaccination coverage, vaccination efficacy, and survival rates of different cancer stages. Birth, death, and ageing rates were entered using a lookup function to accommodate the changing population dynamics.

Altogether, 91 parameters were used in the model. Appendix A in S1 File summarises all main parameters with the evidence source. The crucial parameters comply with the HPV-FRAME Standards of modelling HPV vaccination and cervical cancer screening [40] (Appendix B in S2 File). The model was simulated for 100 years from 1990, as the national screening program was launched in 1998 in Malaysia, to 2090, to allow enough time for elimination to be included in the simulation. The time step was set at one year, and calibration was done in years.

**2.2.5 Model calibration and validation.** The developed model was presented and discussed with two obstetrics gynaecologists and one academician who validated that the disease and intervention structures were accurate through four separate sessions between September 2022 and April 2023. Subsequently, structural and behavioural testing were conducted to validate the model. For structural validation, the unit-checking function in Vensim confirmed that the units used throughout the model were error-free [15], indicating that the equations for all paths were correctly specified according to the variables involved. Next, the extreme condition test was performed, whereby scenarios where no vaccination and no screening were performed were simulated. The model demonstrated a continuous increase in cervical cancer incidences as the population grew, confirming the expected behaviour.

The model was calibrated and validated against several data: Malaysian female population data, national HPV infection prevalence data, number of screening tests performed, national age-standardised cervical cancer incidence rates, national cervical cancer incidences, and global estimation of cervical cancer elimination. To measure the closeness between the simulated trend and actual data, the statistical method Mean Absolute Percent Error (MAPE) was calculated using the following formula in Eq 7:

$$Mean\ Absolute\ Percent\ Error\ (MAPE) = \frac{1}{n}\sum_{t=1}^{n}\left|\frac{S_t - A_t}{A_t}\right| \qquad (Eq\ 7)$$

A value of less than or equal to ten per cent ($\leq 10\%$) was considered a good fit [41,42]. Fig 3 compares simulated behaviours of cancer incidence cases and rates with the historical data. Both parameters had $\leq 10\%$ of the mean absolute percentage errors with the Malaysian data, with behavioural progression akin to actual behaviours. Differences in the fluctuation patterns were most likely due to the considerable yearly variations in the actual data. In contrast, most parameters in the model were either held constant or changed in a linear or stepwise pattern. The validation of other main parameters, including population growth, the prevalence of HPV infection, the number of screening tests performed, and the distribution of cervical cancer by age groups, are reported in Appendix C in S3 File.

## 2.3 Evaluating the impact of different screening transition scenarios

**2.3.1 Establishing the scenarios.** Once the model development, calibration and validation processes were completed, the model was considered stable to simulate different policy scenarios. Continuous and iterative expert engagement sessions were conducted to establish several scenarios to be projected by the model. Four sessions were conducted between September 2022 and April 2023 with seven experts (two public health specialists, one gynaecologist, two medical officers, and two nurses). Considering the capacity and potential, the scenarios were set based on current screening performance and expected screening expansion deemed

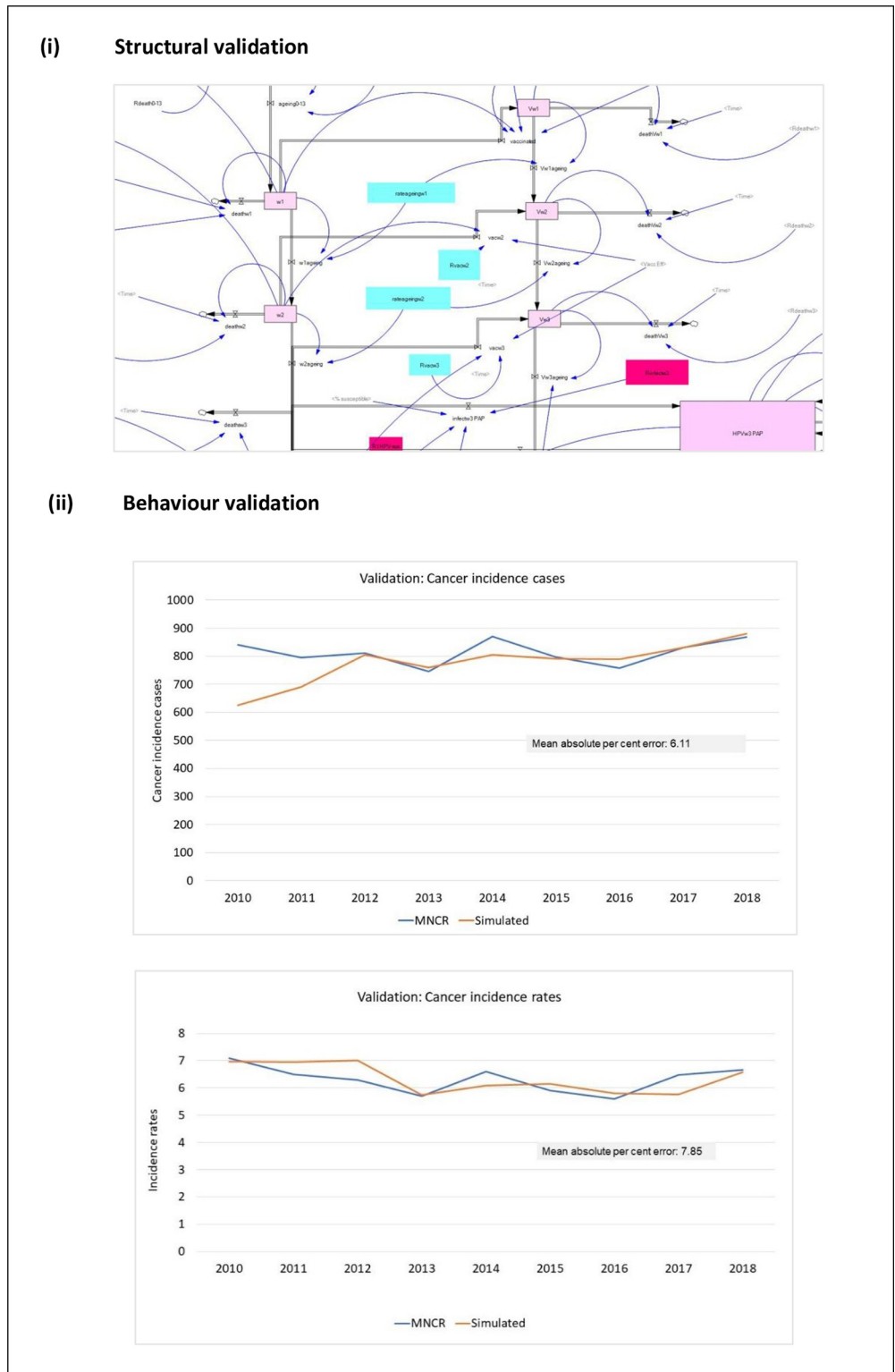

**Fig 3. Structural and behaviour validations of the main parameters in the model.** Note: MNCR = Malaysian National Cancer Registry.

achievable for Malaysia. The Action Plan Towards the Elimination of Cervical Cancer in Malaysia 2021–2030 guided the engagement sessions [6]. The scenarios were divided into two sections: first, to look at the impact of vaccination, followed by the effects of screening.

**2.3.2 Assessing the impact of HPV vaccination.**   To evaluate the impact of vaccination in the absence of screening, the year of elimination was projected for the following scenario: Annual HPV vaccination of girls aged 13 at a population coverage of 90% compared to 70%, all without implementing screening.

To examine the implications of the absence of HPV vaccination, the simulation for the following scenario was conducted: LBC screening at 25% coverage versus 70% coverage, with no HPV vaccination.

To investigate the combined effects of HPV vaccination and screening, the simulation for the following scenario was conducted: Annual HPV vaccination of girls aged 13 at 90% population coverage and LBC screening at 25% coverage. This scenario also served as the baseline in subsequent assessment of the impact of screening.

**2.3.3 Assessing the impact of screening.**   Subsequently, four scenarios incorporating the targeted year for a full transition from LBC to HPV testing, along with an increase in screening coverage, were simulated as follows:

- In Scenario A, the transition from LBC to five-yearly HPV testing occurred gradually till a complete transition was achieved in 2030, with the screening coverage retained at 25% from 2023 onwards.

- In Scenario B, the transition from LBC to five-yearly HPV testing occurred gradually till a complete transition was achieved in 2030, with the screening coverage now increased and retained at 40% in 2030.

- In Scenario C, the transition from LBC to five-yearly HPV testing occurred gradually till a complete transition was achieved in 2030, with the screening coverage increased up to and retained at 70% in 2040.

- In Scenario D, the transition from LBC to HPV testing occurred gradually till a complete transition was achieved in 2030, with the screening coverage now increased up to and retained at 70% in 2040. In this scenario, HPV testing was done twice in a lifetime, first by the age of 40 and second by the age of 50, in concordance with WHO recommendations [1]. The ages for the two screening episodes were adjusted to accommodate the wider age groups screened in Malaysia between 30–65 years.

The screening approach was assumed to be sustained at the same coverage following elimination in all scenarios. The vaccination rate was held constant at a yearly 90% population coverage. In addition, an ideal scenario with 100% HPV vaccination coverage and a high screening rate of 90% was simulated.

Two additional simulations were added to each scenario, projecting:

- the delay in transitioning, whereby the full transition from LBC to HPV was delayed to 2040 and

- the delay in screening coverage increments, whereby achieving 40% and 70% coverage was delayed to 2040 and 2050.

As listed in Table 1, multiple simulated outcomes were examined across scenarios. It included a baseline setting, which mirrors the screening approach through LBC alone at a screening coverage retained at 25%. This baseline configuration served as a reference point, reflecting Malaysia's screening practices before the introduction of HPV testing in 2019. The

**Table 1. Description of scenarios and outcomes simulated.**

| Scenario | Description | Outcomes simulated |
|---|---|---|
| Baseline | Screening for cervical cancer was only done through liquid-based cytology (LBC) three-yearly, with the screening coverage held constant at 25%. | • Year of elimination–the first year of age-standardised cervical cancer incidence, calculated using the Segi-Doll world population standardisation method [43,44], constantly remained below four per 100,000 women.<br>• Number of screening tests needed to be performed from the year 2023 till elimination is achieved.<br>• Total screening costs for each scenario from 2023 till elimination is achieved.<br>•Total screening costs for each scenario from 2023 till 2070 to account for the expenses in sustaining the approach beyond elimination.<br>•Cervical cancer incidence avoided–the number of new cervical cases avoided between 2023 and 2070, compared to the baseline where LBC was continued. |
| Scenario A | The transition from LBC to HPV test occurred gradually until 100% transition was achieved in 2030, AND the screening coverage was retained at 25% from 2023 onwards. | |
| Scenario B | The transition from LBC to HPV tests occurs gradually until 100% transition was achieved in 2030, AND the screening coverage will increase gradually from 25% in 2023 to 40% in 2030. | |
| Scenario C | The transition from LBC to a five-yearly HPV test occurs gradually until a 100% transition was achieved in 2030, AND the screening coverage increased gradually from 25% in 2030 to 40% in 2030 and 70% in 2040. | |
| Scenario D | The transition from LBC to twice-in-a-lifetime HPV tests occurs gradually until a 100% transition was achieved in 2030, AND the screening coverage increases gradually from 25% in 2030 to 40% in 2030 and 70% in 2040. HPV testing was done twice, first by the age of 40 and second by the age of 50. | |

baseline scenario was also employed as a comparative benchmark against all transitional scenarios, allowing assessment of the changes brought about by transitioning to HPV testing instead of persisting with LBC alone.

**2.3.2 Simulating alternative scenarios in screening practice.** To address several uncertainties, a simulation of the following alternative scenarios to screening practice was performed:

- **A combination of screen-treat for HPV 16/18 and screen-triage-treat pathway for non-HPV 16/18** following a positive HPV testing. Three approaches in Scenario A were used for comparison: i) HPV was done five-yearly HPV screening at up to 40% coverage, ii) HPV was done twice a lifetime, up to 70% coverage, and iii) HPV was done five-yearly at 70% coverage.

- **A lower adherence to follow-up** upon a positive screening test at 50–60% (as opposed to 80–90%). In this approach, 70% of women screened with HPV five-yearly were used for comparison.

In addition to these scenarios, sensitivity analyses were conducted on several crucial parameters, including a range of non-adherence rates to follow-up after a positive screening test, varying HPV infection rates across different age groups, and varying unvaccinated rates. The sensitivity analyses were performed based on the baseline model whereby screening was done via LBC sustained at 25% coverage.

## 2.4 Ethics approval and consent to participate

This study was part of an extensive study titled 'Cervical Cancer Screening: Simulation of Best Practices to Achieve the Target' with trial registration number NMRR ID-22-00187-DJU. The study was approved by the Medical Research and Ethics Committee (MREC), Ministry of

Health (KKM, NIHSEC, 22-00187-DJU (1)). The study was conducted according to Good Clinical Practice guidelines and the Declaration of Helsinki. Informed consent was obtained from all approached experts through written documentation. Permission from all data holders was obtained. All identifiers were removed, and data were anonymised before data analysis.

# Results

## 3.1 Impact of HPV vaccination

At a yearly 90% vaccination coverage of girls aged 13 in the population, elimination was projected to be achieved in 2070 (Fig 4, Diagram A). However, if this coverage was reduced to 70%, considering the herd immunity, the model projected the elimination would be delayed by more than 20 years, whereby the age-standardised incidence rate would continue to fluctuate above four per 100,000 women population by 2090 (Fig 4, Diagram A). Furthermore, without vaccination, screening efforts through the LBC approach alone could not achieve and sustain elimination, even if the screening rate increased to 70% of the population (Fig 4, Diagram B). With the combination of high coverage vaccination (90% yearly) and screening through LBC every three years at 25% coverage, the model projected the elimination will be in 2070, the same year of elimination through high coverage HPV vaccination alone (Fig 4, Diagram C).

## 3.2 Impact of screening

**3.2.1 Impact towards the year of cervical cancer elimination.** We analysed four scenarios compared with a baseline scenario and the additional effect of delays in transition and increasing screening coverage. In Scenario A (Fig 5, Diagram A), the model projected a complete transition from three-yearly LBC to five-yearly HPV testing by 2030, with the screening coverage retained at 25% would accelerate the elimination of cervical cancer in Malaysia by two years as opposed to the baseline (2068 vs. 2070). A ten-year delay in transitioning from LBC to HPV testing would delay the elimination by one year (2069 vs. 2068).

In Scenario B (Fig 5, Diagram B), a complete transition from a three-yearly LBC to a five-yearly HPV test combined with increased screening coverage to 40% would accelerate the elimination by 14 years compared to the baseline (2056 vs. 2070). However, a ten-year delay in transitioning from LBC to HPV testing would delay the elimination by 13 years (2069 vs. 2056). In contrast, a delay in increasing the screening coverage from 25% to 40% would result in no delay of elimination.

In Scenario C (Fig 5, Diagram C), a complete transition from three-yearly LBC to five-yearly HPV testing combined with increased screening coverage to 70% would accelerate the elimination by 18 years compared to the baseline (2052 vs. 2070). A delay in in transitioning from LBC to HPV testing would result in delaying the elimination by four years (2056 vs. 2052). In contrast, a delay in increasing the screening coverage from 25% to 40% would result in a three-year delay (2055 vs. 2052).

In Scenario D (Fig 5, Diagram D), a complete transition from three-yearly LBC to twice-in-a-lifetime HPV testing in combination with increased screening coverage to 70% would accelerate the elimination by 14 years as opposed to the baseline (2056 vs. 2070). A delay in transitioning from LBC to HPV testing would delay the elimination by 12 years (2068 vs. 2056). In contrast, a delay in increasing the screening coverage to 70% would result in no elimination delay.

**3.2.2 Impact towards the costs, number of screening tests needed and cancer avoidance.** This study extended the scenario analysis by projecting the impact towards different outcomes. As shown in Table 2, the total costs required to perform the screening until each

(A) Scenario: Impact of vaccination, in the absence of screening

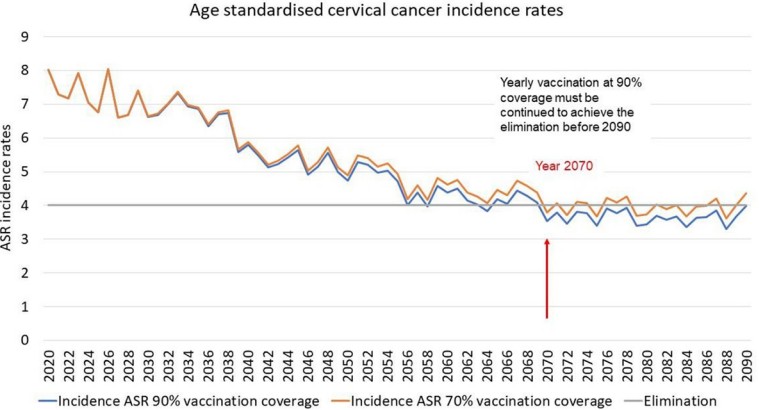

(B) Scenario: Impact of the absence of vaccination, with screening continued

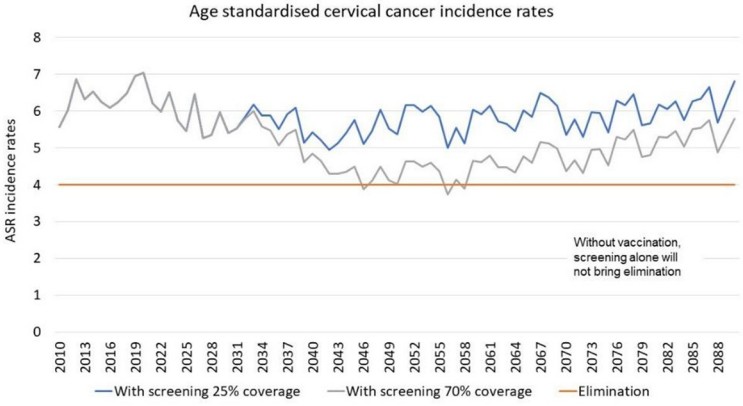

(C) BASELINE: Impact of combination screening (25% screened with LBC every 3 year) and vaccination

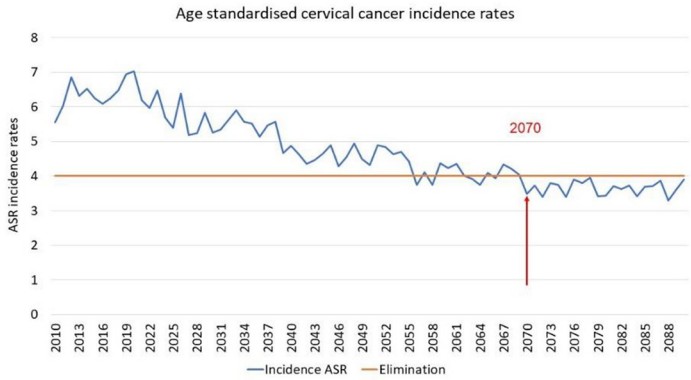

*ASR: Age-standardised cervical cancer incidence rate*

**Fig 4. Impact of vaccination on year of elimination.**

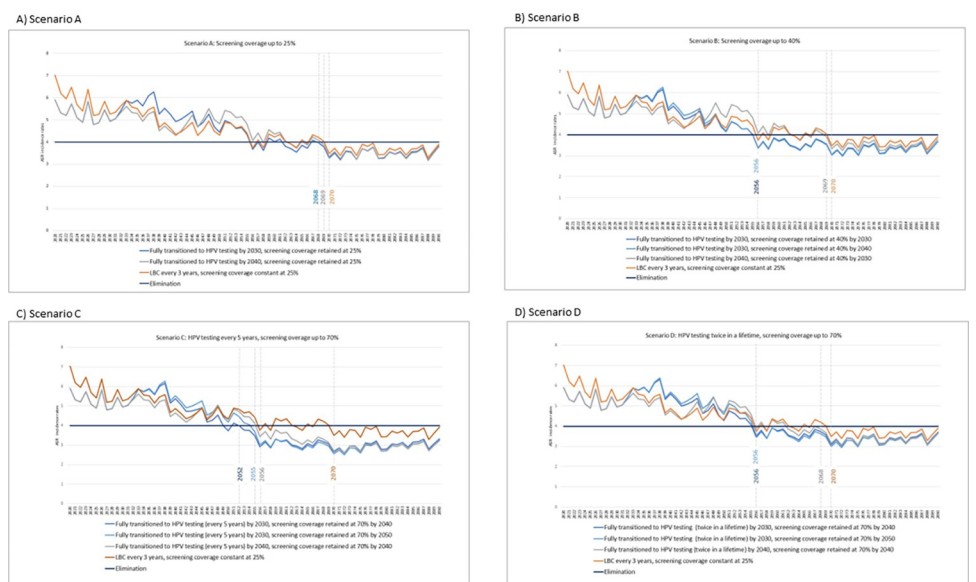

**Fig 5. The simulated year of elimination based on different transition and screening expansion scenarios.**

scenario's respective elimination year were either similar or cheaper in the baseline (MYR 1.39 billion, equivalent to USD 315 million) than all other scenarios. The exception was when HPV testing was done twice in a lifetime at 70% coverage (MYR 1.13 billion, equivalent to USD 257 million), which was projected to save MYR 254 million or USD 58 million. In addition, the total costs for sustaining each approach until 2070 were higher in all approaches than the cost of maintaining the baseline.

Despite the increased cost, the number of tests necessary to perform, calculated from 2023 to the respective year of elimination, was lower in all approaches compared to the baseline, except if HPV testing was done five-yearly and at 70% coverage. This will require, on average, an additional 131,505 (26%) tests per year instead of the baseline (Table 2). In contrast, HPV testing increased up to 70% coverage twice in a lifetime would require 144,868 fewer tests than baseline to achieve elimination.

All approaches would retain a similar number or avoid new cervical cancer cases instead of baseline. The most prominent avoidance was seen with HPV five-yearly combined with 70% screening coverage, leading to a 5,987 lesser incidence between 2023 and 2070, as opposed to if baseline were to continue. Although fewer, HPV twice in a lifetime would reduce the incidence compared to the baseline by 1070 cases by 2070.

## 3.3 Alternative scenarios for screening practice

Alternative scenarios were simulated to address uncertainties in the model. First, the combination of screen-treat for HPV 16/18 and screen-triage-treat pathway for non-HPV 16/18 following positive HPV testing was simulated and shown in Fig 6. At 40% screening coverage, it showed a five-year delay compared to if done entirely through a screen-treat approach when the HPV test was done five-yearly. A five-year delay was also projected through the screen-triage-treat procedure when the HPV test was done twice a lifetime, with up to 70% coverage. However, at 70% screening coverage, no impact towards the year of elimination was observed when the HPV test was done five-yearly.

**Table 2. The detailed costs, screening tests needed to perform, and differences in cervical cancer incidence based on different approaches.**

| Scenarios | Year of elimination | Test needed to perform /year from 2023—elimination | The difference in the number of tests needed to perform /per year | Total screening cost (MYR) from 2023—elimination | Total screening cost difference (MYR) from 2023—elimination | Total screening cost (MYR) from 2023–2070 | Total cancer incidence (2030–2070) | Difference in cancer incidence (2030–2070) |
|---|---|---|---|---|---|---|---|---|
| Baseline: LBC is done 3-yearly at 25% coverage | 2070 | 502,492 | ref | 1,385,145,144 | ref | 1,385,145,144 | 55,518 | ref |
| A: Full transition to five-yearly HPV test in 2030, screening coverage constant at 25% | 2068 | 316,589 | -185,903 | 1,381,646,118 | -3,499,026 | 1,436,149,883 | 55,922 | 404 (+9 per year) * |
| B: Full transition five-yearly HPV test in 2030; screening coverage increases to 40% in 2030 | 2056 | 476,318 | -26,174 | 1,533,054,735 | 147,909,591 | 2,154,536,500 | 53,249 | -2,270 (48 per year) * |
| C: Full transition to five-yearly HPV test in 2030; screening coverage increases to 40% in 2030 and 70% by 2040 | 2052 | 633,998 | 131,505 | 1,807,426,276 | 422,281,132 | 3,197,827,068 | 49,531 | -5,987 (127 per year) * |
| D: Full transition to HPV twice-in-a-lifetime in 2030; screening coverage increases to 40% in 2030 and 70% by 2040 | 2056 | 357,624 | -144,868 | 1,130,153,726 | -254,991,417 | 1,673,782,301 | 54,448 | -1,070 (23 per year) * |

Note

* these are absolute average numbers across the years. The actual yearly incidence averted varies from year to year across all scenarios.

Next, the projected year of elimination when adherence to follow-up was at a lower rate of 50–60%, as opposed to 80–90%. In the scenario where 70% of women were screened with HPV tests five-yearly, a lower follow-up rate caused a four-year delay in achieving elimination (the year 2056 instead of 2053). The diagrammatic representation is shown in Appendix D in the S4 File. The results of the sensitivity analyses are reported in Appendix E in the S5 File.

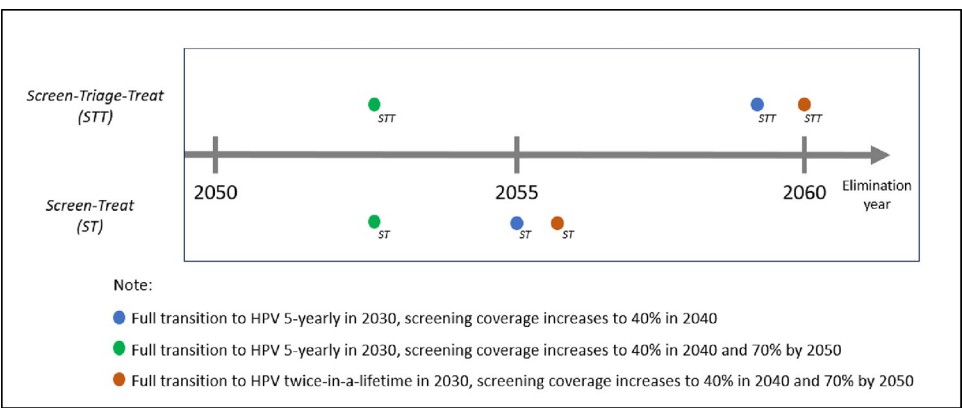

**Fig 6. Projection of cervical cancer elimination comparing the screen-treat and combination with the screen-triage-treat approach.**

## Discussion

This study simulated the transition from LBC to HPV testing as the primary cervical cancer screening, in combination with increments in the screening coverage in Malaysia. The impact of HPV vaccination was also explored. The results showed that yearly population vaccination at 90% coverage is crucial for elimination. A significant acceleration by 18 years in eliminating cervical cancer and avoidance of about 6000 new cancer cases by 2070 in Malaysia are expected through a combination of high vaccination coverage, transitioning fully to five-yearly HPV testing by 2030, and increased screening uptake by up to 70% by 2040. By the same principles of approach, screening women with HPV twice before age 50 will eliminate cervical cancer 14 years earlier and prevent about 1000 incidence cases by 2070. While the total costs of screening in achieving elimination would increase compared to the baseline, all approaches would accelerate elimination by requiring fewer screening tests.

The model projected that 90% yearly population coverage in HPV vaccination would lead to cervical cancer elimination in 2070. This corresponds with the global projection that high HPV vaccination coverage would bring elimination before the end of the century in most world regions [45]. In contrast, without screening, reducing HPV vaccination to 70% of population coverage would delay the elimination by over 20 years. Considering the effect of herd immunity, the acceptable range shown in the meta-analysis to achieve 100% protection from high-risk HPV 16/18 infection by vaccinating girls only is around 80–85% [32]. This reflects the current practice in Malaysia, whereby the population coverage ranges from 83–91% between 2010 to 2016 [3]. However, there is a need to explore the population coverage by geographical variations in Malaysia since national-level monitoring often masks geographical inequalities. Nevertheless, a policy to sustain the high yearly vaccination coverage must be in place, as a drop in coverage would lead to considerable impact. The recent move by MOH Malaysia to adopt a single-dose vaccine following WHO's recommendation represented a significant step towards sustaining the high coverage [46,47]. However, a readjustment of the model may be needed once more evidence of the difference in efficacy and duration of protection of a single-dose vaccine is attained.

The model simulated an 18 and 14-year earlier elimination when high screening coverage of up to 70% of women when HPV testing was done on a five-yearly basis or at least twice before the age of 50 compared to performing an LBC. This corresponds to the global projection that predicts an 11–31 years elimination acceleration brought by effective screening using HPV tests and high HPV vaccination coverage [45]. It also supports the previous model showing that the combination would bring an 11–20 years expedited elimination in Malaysia [14]. Furthermore, conversion to HPV tests would prevent a substantial amount of cervical cancer incidences in the next few decades than if LBC were to continue. The findings align with simulations conducted elsewhere, demonstrating the benefits of transitioning from LBC to HPV testing. This transition not only reduces the incidence of cervical cancer but also minimises the resources required for implementing the screening programme [48,49]. For example, a simulation conducted for Australian women found that transitioning from LBC to HPV testing as the primary screening method would reduce the risk of cancer development by 33% for unvaccinated women and by 22% for vaccinated women [49]. Another simulation demonstrated that the transition initially caused a temporary increase in early-stage precancerous incidences. Still, these rates steadily declined, reducing precancerous incidences by 40–44% and invasive cancer incidences by 42–51% [50]. Similarly, a simulation for New Zealand women showed that transitioning to HPV testing would reduce cervical cancer incidence by 32% and mortality rates by 25% [48]. It is worth noting that the number of prevented cervical cancer cases, as evidenced in this study, is somewhat lower than global statistics and prior

Malaysian models. This discrepancy arises from the study approach, where a baseline scenario assuming continued high coverage of HPV vaccination and LBC screening in the absence of HPV testing was used rather than comparing against scenarios without vaccination and screening as previously modelled [14,45].

The simulation highlights that at a low screening coverage of 25%, sustaining screening through LBC had no impact towards the year of elimination compared to when high population coverage with HPV vaccination alone was implemented. Efforts to expedite the transition process are necessary since a delay in achieving the complete transition is simulated to cause up to thirteen years of delay in achieving elimination. In contrast, when a full transition from LBC to HPV test is attained by 2030, a 10-year delay in increasing the screening coverage only led to a maximum of a three-year delay in elimination. This shows that a delay in the complete transition from LBC to HPV may have a more significant influence towards the elimination year than a delay in increasing the screening coverage and should be emphasised. This is likely from the higher efficacy of HPV testing in detecting abnormality at a much earlier stage than LBC [10].

This finding is crucial amidst Malaysia's progress towards fully converting to HPV tests by 2030 [6]. As of 2022, 20% of all screening among Malaysian women were done through HPV tests, with the remaining conducted through LBC [51], with the total screening coverage remaining at around 16–25% of at-risk women [4,5]. The findings suggest that prioritising a complete transition from LBC to HPV tests could be a more favourable approach while overcoming the challenges of encouraging more women to participate in the screening process. An expedited transition to the HPV test would be beneficial since the HPV test only needs to be repeated five years following a negative result, as opposed to three years for LBC [52]. This results in fewer required screening tests. Furthermore, HPV tests can be performed through self-sampling, opening up opportunities to expand the outreach for women in the community, perhaps by considering decentralised and at-home tests via postage of screening kits, as piloted or implemented in various settings [11,53,54]. Nevertheless, it is essential to complement these strategies with effective monitoring, preferably through implementing a digitised national screening registry, to prevent redundant screenings of the same individuals and avoid missing women who may not be responding to screening invitations and subsequent follow-ups. Indeed, the alternative scenario analysis shows that if the follow-up rate drops from 80–90% to around 50–60%, the elimination will be further delayed by several years. Hence, within the context of Malaysian health systems, where the screening method has been opportunistic, lacking both a digital screening registry and an efficient follow-up system, maintaining a five-yearly HPV test, despite its increased cost, could emerge as a more effective screening strategy compared to the twice-in-a-lifetime approach.

Owing to the increased expense associated with conducting HPV tests, the total screening costs for all scenarios were higher than the baseline, except in cases where HPV testing was performed twice over a lifetime. The higher cost of performing HPV is due to higher laboratory processing requirements to run the polymerase chain reaction (PCR) test [36], and efforts to reduce the price gap are necessary. One of the ongoing efforts involves outsourcing the laboratory processes, which have been rolled out in phases across multiple regions in Malaysia since 2019 [6]. A cost-effectiveness study is recommended to analyse the relative cost implications based on the health outcomes, which is beyond the purview of the current study.

The study adopts system dynamic simulation, a technique allowing complex system analysis in projecting the impact of transitioning and expanding cervical cancer screening. The simulated parameters were within acceptable error ranges with actual data, indicating that the model was suitable for scenario analysis. However, the underlying assumptions limit the model and may behave differently under different conditions. The model also assumed that

most parameters remain constant throughout the simulation when, in reality, the population's behaviour and health systems often evolve. Furthermore, the data used for the simulation was averaged across the whole country, when different regions, especially the population of the eastern areas of Malaysia, have different demographics and unique challenges. Remodelling based on region-specific data is necessary once more regional data becomes available. The population growth also did not account for the migration rate. Despite the limitations, the study is one of the few simulations projecting the transition from a more traditional approach to cervical cancer screening to a more effective method, coupled with a gradual increment of the screening coverage. Coming from a southeastern Asian country of a middle-income background, the findings are suitable as a reference for countries with similar demographic and health system challenges.

## Conclusions

With inputs from local experts and policymakers, the study highlights the potential use of a simulation model to offer pragmatic strategies for policy considerations in the national cervical cancer program, leveraging two crucial areas- time to complete transition and screening coverage- thus expediting the eradication of cervical cancer. By 2070, this proactive approach could accelerate elimination by up to 18 years and prevent nearly 6,000 cancer cases, starkly contrasting with the baseline strategy relying solely on traditional LBC. A delay in achieving the full transition would substantially prolong the elimination by up to 13 years, with a greater impact than a delay in increasing the screening coverage. The findings thus support the current MOH policy move to prioritise the full conversion of the screening to HPV test by 2030 while addressing the challenges associated with reaching more women to attend the screening.

## Supporting information

**S1 File. Appendix A: List of parameters used.**
(PDF)

**S2 File. Appendix B: HPV-FRAME standard.**
(PDF)

**S3 File. Appendix C: Validity of model.**
(PDF)

**S4 File. Appendix D: Alternative scenarios.**
(PDF)

**S5 File. Appendix E: Sensitivity analysis.**
(PDF)

## Acknowledgments

The authors would like to declare our gratitude to the Director-General of Health Malaysia for his permission to publish this paper. All authors thank the Family Health Development Division, MOH, and the National Cancer Registry Malaysia for permission and data provision for this study.

## Author Contributions

**Conceptualization:** Nur Zahirah Balqis-Ali, Jailani Anis-Syakira, Weng Hong Fun, Sondi Sararaks.

**Data curation:** Nur Zahirah Balqis-Ali, Jailani Anis-Syakira, Zakiah Mohd Said, Shazimah Abdul Samad, Siti Norbayah Yusof, Nur Adila Daud.

**Formal analysis:** Nur Zahirah Balqis-Ali, Jailani Anis-Syakira, Weng Hong Fun.

**Funding acquisition:** Nur Zahirah Balqis-Ali, Jailani Anis-Syakira.

**Investigation:** Nur Zahirah Balqis-Ali, Jailani Anis-Syakira, Weng Hong Fun, Zakiah Mohd Said, Shazimah Abdul Samad.

**Methodology:** Nur Zahirah Balqis-Ali, Jailani Anis-Syakira, Weng Hong Fun, Norhaslinda Zainal Abidin, Jafri Zulkepli, Norazura Ahmad, Sondi Sararaks.

**Project administration:** Nur Zahirah Balqis-Ali, Jailani Anis-Syakira, Weng Hong Fun, Sondi Sararaks.

**Resources:** Zakiah Mohd Said, Shazimah Abdul Samad, Siti Norbayah Yusof, Nur Adila Daud.

**Software:** Nur Zahirah Balqis-Ali, Jailani Anis-Syakira, Norhaslinda Zainal Abidin, Jafri Zulkepli, Norazura Ahmad.

**Supervision:** Weng Hong Fun, Zakiah Mohd Said, Shazimah Abdul Samad, Norhaslinda Zainal Abidin, Jafri Zulkepli, Norazura Ahmad, Mohd Norazam Mohd Abas, Chee Meng Yong, Sondi Sararaks.

**Validation:** Nur Zahirah Balqis-Ali, Jailani Anis-Syakira, Weng Hong Fun, Zakiah Mohd Said, Shazimah Abdul Samad, Norhaslinda Zainal Abidin, Jafri Zulkepli, Norazura Ahmad, Mohd Norazam Mohd Abas, Chee Meng Yong, Siti Norbayah Yusof, Nur Adila Daud, Sondi Sararaks.

**Visualization:** Nur Zahirah Balqis-Ali, Jailani Anis-Syakira.

**Writing – original draft:** Nur Zahirah Balqis-Ali, Jailani Anis-Syakira, Weng Hong Fun, Sondi Sararaks.

**Writing – review & editing:** Nur Zahirah Balqis-Ali, Jailani Anis-Syakira, Weng Hong Fun, Zakiah Mohd Said, Shazimah Abdul Samad, Norhaslinda Zainal Abidin, Jafri Zulkepli, Norazura Ahmad, Mohd Norazam Mohd Abas, Chee Meng Yong, Siti Norbayah Yusof, Nur Adila Daud, Sondi Sararaks.

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
