## [Decision Letter · Decision Letter 0]

30 May 2024

Dear Dr. Anis-Syakira,

Thank you for submitting your manuscript  titled " PONE-D-24-07127:From model to action: Simulating impact of transitioning from liquid-based cytology to HPV-based screening test on cervical cancer elimination" to PLOS ONE. After careful consideration, we feel that it has merit but does not fully meet PLOS ONE’s publication criteria as it currently stands. Therefore, we invite you to submit a revised version of the manuscript that addresses the points raised during the review process.

Please include the following items when submitting your revised manuscript:A rebuttal letter that responds to each point raised by the academic editor and reviewer(s). You should upload this letter as a separate file labeled 'Response to Reviewers'.A marked-up copy of your manuscript that highlights changes made to the original version. You should upload this as a separate file labeled 'Revised Manuscript with Track Changes'.An unmarked version of your revised paper without tracked changes. You should upload this as a separate file labeled 'Manuscript'.

We look forward to receiving your revised manuscript by Jul 14 2024 11:59PM.

Kind regards,

Yitagesu Habtu Aweke, Ph.D

Academic Editor

PLOS ONE

**Additional Editor Comments :**

In addition to reviewers' comment,  I suggest you to re-check the impact models, and check  your label  for the subheadings 

**Reviewers' comments:**

Reviewer's Responses to Questions

**Comments to the Author**

1. Is the manuscript technically sound, and do the data support the conclusions?

Reviewer #1: Yes

Reviewer #2: Yes

Reviewer #3: Partly

2. Has the statistical analysis been performed appropriately and rigorously? 

Reviewer #1: Yes

Reviewer #2: Yes

Reviewer #3: I Don't Know

3. Have the authors made all data underlying the findings in their manuscript fully available?

Reviewer #1: Yes

Reviewer #2: Yes

Reviewer #3: Yes

4. Is the manuscript presented in an intelligible fashion and written in standard English?

Reviewer #1: Yes

Reviewer #2: Yes

Reviewer #3: Yes

5. Review Comments to the Author

Reviewer #1: The authors have presented a well-conducted and important study. In addition to the data presented, it will be useful for the authors to highlight more studies done in other countries that have successfully made the transition from LBC to HPV testing as the primary cervical cancer screening.

Reviewer #2: This study analyzed the impact of HPV vaccine coverage, transition of screening modalities, and screening coverage on the goal of eliminating cervical cancer. It quantitatively calculated the years to achieve the goal of cervical cancer elimination under different resource allocation scenarios, along with cost estimation. The study confirmed the importance of transitioning screening modalities comprehensively to HPV testing, providing a solid scientific basis for prioritizing policy support for HPV testing. However, several areas require further attention and improvement.

1.What would happen if the vaccine coverage rate increased to 95% or 100%? What would be the impact if the screening coverage reached 80% or above? The modeling and simulation study can provide a scientific basis for the government to set and adjust goals.

2.Since the selection of parameters is crucial for the model results, especially for parameters like vaccine efficacy obtained from references in other countries and unpublished parameters from expert consultations, could sensitivity analysis be conducted to enhance the study?

Reviewer #3: Title should be changed since the vaccination is in the model. My suggestion is: Finding the best strategy for cervical cancer elimination-because

In abstract: increasing screening coverage from which %

MYR should be converted in US dollars because of international comparison

In abstract (46th raw): I don't understand how much in MYR is difference between two models- related to results..unclear

English should be more clear (exp. 110-111 raw)

6. PLOS authors have the option to publish the peer review history of their article (what does this mean?). If published, this will include your full peer review and any attached files.

Reviewer #1: No

Reviewer #2: No

Reviewer #3: No

---

## [Author Response · Author response to Decision Letter 0]

11 Jun 2024

Response to editor and reviewers

Editor

In addition to the reviewers' comments, I suggest you re-check the impact models and check your label for the subheadings.

Response: Thank you for your constructive feedback on our manuscript. We sincerely appreciate your encouragement and have diligently addressed the valuable comments the editor and reviewers provided. Regarding the impact models, we have reevaluated the model, results, and interpretations, confirming that the findings are based on the simulations. Additionally, the subheadings have been reviewed.

We acknowledge that PLOS One uses a non-numbering system for subheadings. However, due to the complexity of model development and analysis and multiple layers of subheadings, we believe retaining the current numbering system will enhance readability and facilitate understanding for readers. The font formatting of the subheadings is in accordance with PLOS One’s requirement. 

Reviewer #1: 

The authors have presented a well-conducted and important study. In addition to the data presented, it will be useful for the authors to highlight more studies done in other countries that have successfully made the transition from LBC to HPV testing as the primary cervical cancer screening. 

Response: Thank you for your constructive feedback on our manuscript. We sincerely appreciate your encouragement and have diligently addressed the valuable comments the editor and reviewers provided. In response to your suggestion, we have expanded the discussion section to include more studies that test the impact of transitioning from LBC to HPV test in screening for cervical cancer, along with their findings. 

Change: Page 24 - 25, Line 571 - 581, clean version

The findings align with simulations conducted elsewhere, demonstrating the benefits of transitioning from LBC to HPV testing. This transition not only reduces the incidence of cervical cancer but also minimises the resources required for implementing the screening programme [51, 52]. For example, a simulation conducted for Australian women found that transitioning from LBC to HPV testing as the primary screening method would reduce the risk of cancer development by 33% for unvaccinated women and by 22% for vaccinated women [52]. Another simulation demonstrated that the transition initially caused a temporary increase in early-stage precancerous incidences. Still, these rates steadily declined, reducing precancerous incidences by 40-44% and invasive cancer incidences by 42-51% [53]. Similarly, a simulation for New Zealand women showed that transitioning to HPV testing would reduce cervical cancer incidence by 32% and mortality rates by 25% [51].

Reviewer #2: 

This study analyzed the impact of HPV vaccine coverage, transition of screening modalities, and screening coverage on the goal of eliminating cervical cancer. It quantitatively calculated the years to achieve the goal of cervical cancer elimination under different resource allocation scenarios, along with cost estimation. The study confirmed the importance of transitioning screening modalities comprehensively to HPV testing, providing a solid scientific basis for prioritizing policy support for HPV testing. We deeply respect your comments and acknowledge that several areas require further attention and improvement. 

1. What would happen if the vaccine coverage rate increased to 95% or 100%? What would be the impact if the screening coverage reached 80% or above? The modelling and simulation study can provide a scientific basis for the government to set and adjust goals. 

Response: Thank you for your constructive feedback on our manuscript. We sincerely appreciate your encouragement and have diligently addressed the valuable comments the editor and reviewers provided. In relation to the next comment, we expanded the simulation to include sensitivity analyses. In the sensitivity analyses, we have incorporated varying vaccination rates as one of the parameters being tested. The scenario whereby the HPV vaccination rate reached 100% coverage was simulated as part of the sensitivity testing. Given the extensive methods and results sections in the manuscript, we have included the results of the sensitivity analyses in the appendix and referenced them in the results section. We have also tested a higher screening coverage; however, in the situation where the vaccination rate is sustained at a high level (90%), the impact of higher screening coverage is minimal. We decided to retain the screening rate at 70%, as targeted by the Ministry of Health’s policymakers. 

Change: 

Method: Page 16, Line 435 - 439, clean version

In addition to these scenarios, sensitivity analyses were conducted on several crucial parameters, including a range of non-adherence rates to follow-up after a positive screening test, varying HPV infection rates across different age groups, and varying unvaccinated rates. The sensitivity analyses were performed based on the baseline model whereby screening was done via LBC sustained at 25% coverage. 

Results: Appendix E in File S5

2. Since the selection of parameters is crucial for the model results, especially for parameters like vaccine efficacy obtained from references in other countries and unpublished parameters from expert consultations, could sensitivity analysis be conducted to enhance the study? 

Response: Thank you for your suggestion. We recognise the importance of conducting sensitivity analyses in simulation models. As mentioned in response to the previous comment, we have expanded the simulation to include sensitivity analyses.

We also acknowledge the reviewer’s suggestion to include vaccine efficacy in the sensitivity analysis. However, vaccine efficacy varies based on the recipient group (e.g., it decreases when given to non-sexually naïve women) and different HPV types. Currently, the simulation model adheres to the Malaysian vaccination practice whereby the vaccine is administered to 13-year-old girls, presumed to be sexually naïve, and therefore 100% effective against high-risk HPV types. This assumption, which limits the simulation, is documented in the manuscript. With this information and assumption, vaccination efficacy is not tested in the sensitivity analysis. Instead, the vaccination rate is tested in the sensitivity analysis. We appreciate your understanding and hope this decision maintains the clarity and precision of the manuscript.

Reviewer #3: 

The title should be changed since the vaccination is in the model. My suggestion is: Finding the best strategy for cervical cancer elimination-because 

Response: Thank you for your constructive feedback on our manuscript. We sincerely appreciate your encouragement and have diligently addressed the valuable comments the editor and reviewers provided. We have revised the title accordingly to incorporate HPV vaccination.

Change: Title: Achieving cervical cancer elimination: The simulated impacts of HPV vaccination and transitioning from liquid-based cytology to HPV-based screening test

In abstract: increasing screening coverage from which % 

Response: Thank you for your suggestion. The sentence has now incorporated a baseline screening coverage.

Change: Page 2, Line 35 - 37, clean version

The Ministry of Health Malaysia aims to fully replace liquid-based cytology (LBC) with Human Papillomavirus (HPV) tests and increase the screening coverage from a baseline of 25% to 40% by 2023, followed by a 10% yearly increment until 70% coverage.

MYR should be converted in US dollars because of international comparison 

Response: Thank you for your suggestion. The MYR values in the abstract are now accompanied by the equivalent amount in USD.

Change: Improved abstract

In abstract (46th raw): I don't understand how much in MYR is difference between two models- related to results.

Response: Thank you for your comment. The difference in the cost between LBC and HPV tests is mentioned in the methodology section, under the subheading 2.2.2: Model assumptions, as follows (Page 9, Line 244 - 250):

6. Screening costs: MYR 58.66 (USD13.33) per LBC test and MYR 102.86 (USD23.38) per HPV test done through in-house laboratory processing. MYR 84.55 (USD19.22) per HPV test done through outsourcing the laboratory processing was done as an alternative scenario. The details in the cost calculation are reported elsewhere [39]. The monetary figures in this study were denominated in the local currency unit (MYR) and converted to USD, utilising the World Bank's official exchange rate for 2022 (1 USD equivalent to 4.4 MYR) [40]. Constant screening costs without discounted rates were applied in the simulation. 

The screening cost via HPV test is higher than the traditional LBC test. That is why the total cost for HPV tests is higher than LBC, despite its superiority in detecting cervical cancer. These findings are interpreted and discussed in the discussion section as follows (Page 26, Line 618 - 626):

Owing to the increased expense associated with conducting HPV tests, the total screening costs for all scenarios were higher than the baseline, except in cases where HPV testing was performed twice over a lifetime. The higher cost of performing HPV is due to higher laboratory processing requirements to run the polymerase chain reaction (PCR) test [56], and efforts to reduce the price gap are necessary. One of the ongoing efforts involves outsourcing the laboratory processes, which have been rolled out in phases across multiple regions in Malaysia since 2019 [6]. A cost-effectiveness study is recommended to analyse the relative cost implications based on the health outcomes, which is beyond the purview of the current study. 

We hope this clarifies the difference between the two approaches. 

unclear English should be clearer (exp. 110-111 raw)

Response: Thank you for your comment. We have improved the sentence structure as advised.

Change: Page 4, Line 111 - 113, clean version

Although many dynamic models of cervical cancer screening programs have been developed and tested [14, 16-18], each model was created to address specific objectives and was applied to different health system contexts.

Journal Requirements:

Response: The formatting has been checked, complying with PLOS ONE’s style requirement.

Not applicable.

3. We note that you have indicated that there are restrictions to data sharing for this study. For studies involving human research participant data or other sensitive data, we encourage authors to share de-identified or anonymized data. However, when data cannot be publicly shared for ethical reasons, we allow authors to make their datasets available upon request. For information on unacceptable data access restrictions, please see http://journals.plos.org/plosone/s/data-availability#loc-unacceptable-dataaccess-restrictions.

Response: The data availability section has been updated. 

Change: Page 27 – 28, Line 665 – 672, clean version

Availability of data and materials

The dataset supporting this article’s findings belongs to the Family Health Development Division, MOH Malaysia and the National Cancer Registry, Malaysia. However, restrictions apply to the availability of these data, which were used under license for the current study and are not publicly available. Data sharing is subjected to a formal request and approval by data holders and the Director-General of Health, Malaysia. Data are, however, available upon reasonable request from the Principal Investigator (anissyakira.j@moh.gov.my) or through the National Institutes of Health Malaysia-Data Repository System (NIH-DaRS), accessible from https://nihdars.nih.gov.my/. 

Response: The data availability section has been updated. 

Change: Page 27 – 28, Line 665 – 672, clean version

Availability of data and materials

The dataset supporting this article’s findings belongs to the Family Health Development Division, MOH Malaysia and the National Cancer Registry, Malaysia. However, restrictions apply to the availability of these data, which were used under license for the current study and are not publicly available. Data sharing is subjected to a formal request and approval by data holders and the Director-General of Health, Malaysia. Data are, however, available upon reasonable request from the Principal Investigator (anissyakira.j@moh.gov.my) or through the National Institutes of Health Malaysia-Data Repository System (NIH-DaRS), accessible from https://nihdars.nih.gov.my/. 

Response: We have moved the ethics statement into the method section of the manuscript.

---

## [Decision Letter · Decision Letter 1]

7 Jul 2024

PONE-D-24-07127R1Achieving cervical cancer elimination: The simulated impacts of HPV vaccination and transitioning from liquid-based cytology to HPV-based screening testPLOS ONE

Dear Dr. Anis-Syakira,

Thank you for submitting your manuscript to PLOS ONE. After careful consideration, we feel that it has merit but does not fully meet PLOS ONE’s publication criteria as it currently stands. Therefore, we invite you to submit a revised version of the manuscript that addresses the points raised during the review process.

We look forward to receiving your revised manuscript.

Kind regards,

Yitagesu Habtu Aweke, Ph.D

Academic Editor

PLOS ONE

Journal Requirements:

Reviewers' comments:

Reviewer's Responses to Questions

**Comments to the Author**

1. If the authors have adequately addressed your comments raised in a previous round of review and you feel that this manuscript is now acceptable for publication, you may indicate that here to bypass the “Comments to the Author” section, enter your conflict of interest statement in the “Confidential to Editor” section, and submit your "Accept" recommendation.

Reviewer #3: All comments have been addressed

2. Is the manuscript technically sound, and do the data support the conclusions?

Reviewer #3: Yes

3. Has the statistical analysis been performed appropriately and rigorously? 

Reviewer #3: I Don't Know

4. Have the authors made all data underlying the findings in their manuscript fully available?

Reviewer #3: Yes

5. Is the manuscript presented in an intelligible fashion and written in standard English?

Reviewer #3: Yes

6. Review Comments to the Author

Reviewer #3: **With what strategy The Ministry of Health Malaysia aims to fully replace liquid-based cytology (LBC) with Human,  Papillomavirus (HPV) tests and increase the screening coverage from a baseline of 25% to 40% by 2023, followed by a 10% yearly increment until 70% coverage?**

7. PLOS authors have the option to publish the peer review history of their article (what does this mean?). If published, this will include your full peer review and any attached files.

Reviewer #3: No

---

## [Author Response · Author response to Decision Letter 1]

11 Jul 2024

Response to editor and reviewers

Journal Requirements:

Response: Thank you for your constructive feedback. We have reviewed and updated the reference list. We would like to confirm there is no citation of retracted or unpublished materials in the reference list. 

Reviewer #3: 

With what strategy The Ministry of Health Malaysia aims to fully replace liquid-based cytology (LBC) with Human, Papillomavirus (HPV) tests and increase the screening coverage from a baseline of 25% to 40% by 2023, followed by a 10% yearly increment until 70% coverage?

Response: Thank you for your comment. The strategy being proposed by the Ministry of Health Malaysia to replace LBC with HPV tests and increase screening coverage has been added to the introduction session, along with the relevant sources. 

Change: Page 4, Line 95-98, clean version

To achieve this, the MOH plans to increase the yearly procurement of HPV test kits to replace LBC and expand screening through collaborative platforms such as private facilities, workplace screening programs, and nonprofit organizational activities [6, 13].

---

## [Editor Report · Decision Letter 2]

15 Jul 2024

Achieving cervical cancer elimination: The simulated impacts of HPV vaccination and transitioning from liquid-based cytology to HPV-based screening test

PONE-D-24-07127R2

Dear Dr. Jailani Anis-Syakira,

We’re pleased to inform you that your manuscript has been judged scientifically suitable for publication and will be formally accepted for publication once it meets all outstanding technical requirements.

Kind regards,

Yitagesu Habtu Aweke, Ph.D

Academic Editor

PLOS ONE

---

## [Editor Report · Acceptance letter]

17 Jul 2024

PONE-D-24-07127R2 

PLOS ONE

Dear Dr. Anis-Syakira, 

I'm pleased to inform you that your manuscript has been deemed suitable for publication in PLOS ONE. Congratulations! Your manuscript is now being handed over to our production team.

Kind regards, 

on behalf of

PhD Candidate Yitagesu Habtu Aweke 

Academic Editor

PLOS ONE